# Leveraging Constraint Logic Programming for Neural Guided Program Synthesis

**Lisa Zhang[1,2], Gregory Rosenblatt[4], Ethan Fetaya[1,2], Renjie Liao[1,2,3], William E. Byrd[4], Raquel Urtasun[1,2,3] & Richard Zemel[1,2]**
[1]University of Toronto, [2]Vector Institute, [3]Uber ATG, [4]University of Alabama at Birmingham
[1]{lczhang,ethanf,rjliao,urtasun,zemel}@cs.toronto.edu
[4]{gregr,webyrd}@uab.edu

## Abstract

We present a method for solving Programming by Example (PBE) problems that tightly integrates a neural network with a constraint logic programming system called miniKanren. Internally, miniKanren searches for a program that satisfies the recursive constraints imposed by the provided examples. Our Recurrent Neural Network (RNN) model uses these constraints as input to score candidate programs. We show evidence that using our method to guide miniKanren's search is a promising approach to solving PBE problems.

## 1 Introduction

Programming by Example (PBE) is the problem of synthesizing a program specified in terms of input/output examples. State-of-the-art approaches use symbolic techniques developed by the programming languages community (Feser et al., 2015; Albarghouthi et al., 2013; Osera & Zdancewic, 2015), but success in PBE has been limited to small programs in restricted domain specific languages (DSL). These approaches are difficult to scale due to the exponential search space. Recent approaches by the machine learning community attempt to avoid or alleviate the scaling problem. These approaches include differentiable programming (Neelakantan et al., 2016; Reed & de Freitas, 2016), directly synthesizing a program as a sequence or tree (Devlin et al., 2017; Parisotto et al., 2017), and guiding a symbolic search using a neural model (Kalyan et al., 2018; Balog et al., 2017).

We take the latter approach, but take the integration with a symbolic system even further: we use its internal representations as input. The symbolic system used is called miniKanren[1] (Byrd & Friedman, 2006), chosen for its potential to synthesize recursive programs in dynamically typed languages (Byrd et al., 2017). Internally, miniKanren searches for a program that satisfies the recursive constraints (usually called "goals") imposed by the input/output examples. Our model uses these constraints to score candidate programs and guide miniKanren's search.

Our method is promising for several reasons. First, while symbolic systems have performed better, neural guidance can help navigate exponentially large search spaces, leveraging progress made in both communities. Second, symbolic systems exploit the compositionality of synthesis problems: miniKanren's constraints select portions of the input/output examples relevant to a subproblem. This is akin to having a symbolic attention mechanism. Lastly, it is difficult for neural techniques to synthesize programs larger than those seen in training (Parisotto et al., 2017). Guiding a search and using constraints both alleviate this problem. We present some evidence that our approach is able to generalize to programs larger than those seen in training.

## 2 Model

Our approach, along with integration with miniKanren, is summarized in Figure 1.

**Constraint Representation** The symbolic system miniKanren is a constraint logic programming language, equipped with a relational interpreter `evalo` of the target DSL. In our case we use a

---

[1]The name "Kanren" comes from the Japanese word for "relation".

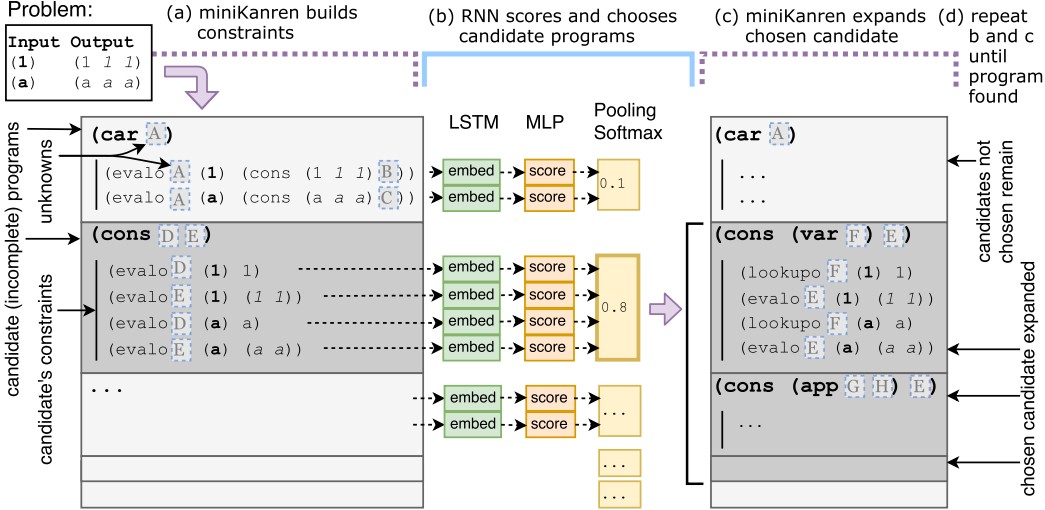

Figure 1: Steps for synthesizing a program that repeats a symbol three times using a subset of Lisp: (a) miniKanren builds constraints representing the search space for the PBE problem; candidate programs contain unknowns, whose values are restricted by constraints; the constraint `(evalo P I O)` is satisfied when an expression `P` with input `I` is evaluated to output `O`; the constraint `(lookupo N E V)` is satisfied when data `E` contains datum `V` at position `N`; in the second candidate, the `evalo` constraints decompose the output into two portions to be synthesized independently; (b) we use an RNN operating on the constraints to score candidates; each constraint is embedded and scored separately, then pooled per candidate; the softmax probability scores determine which candidate to expand; (c) miniKanren expands the chosen candidate `(cons D E)`, so that different completions of unknown `D` are added to the set of candidates; (d) this process of scoring, choosing, and expanding candidates is repeated until a candidate with no unknowns satisfies all constraints imposed by the input/output examples.

dialect of Lisp (Abelson et al., 1996). Here, `evalo` defines a set of recursive constraints (recursive because the constraints may contain other invocations of `evalo`) imposed by the input/output examples. We omit a detailed summary of background work not directly relevant to the machine learning portion of the work, and refer readers to Byrd et al. (2017).

**Search**  The search for a target program is divided into steps. At each step, a candidate incomplete program is chosen and expanded. The default search process used by miniKanren is a biased interleaving search: the search alternates between partial programs, but is biased towards partial programs with more constraints already satisfied. In Byrd et al. (2017), authors present heuristics that improve the search.

**Recurrent Neural Network Model**  We use an RNN with bi-directional Long Short-Term Memory (LSTM) units (Hochreiter & Schmidhuber, 1997) to score candidates. We learn separate RNN weights for each constraint type (`evalo`, `lookupo`, etc). At each step, we embed each constraint using the corresponding LSTM. The embeddings are individually scored using a single scoring function, then the scores are pooled using a combination of mean-pooling and sum-pooling. We softmax over the scores for each candidate program, then make a discrete choice to predict the optimal candidate to expand. In the rest of this work, we refer to our model as RNN-Guided search.

**Training**  We use a small subset of Lisp as our target DSL. This subset consists of `cons`, `car`, `cdr`, along with several constants and function application. We programmatically generate[2] target programs of varying sizes, along with five corresponding input/output pairs. The target program is available during training, and there is a unique optimal candidate that should be selected at each step. We use cross-entropy loss on the softmax probabilities computed across all possible candidate

---

[2]We run the relational interpreter `evalo` "backwards" to generate arbitrary programs using miniKanren.

programs. We use curriculum learning, beginning with problems with shorter target programs and thus fewer synthesis steps, then gradually allowing larger ones. To reduce training time, we use prioritized experience replay (Schaul et al., 2016) and sample mini-batches from a replay buffer. We use scheduled sampling (Bengio et al., 2015) with a linear schedule, to increase exploration and reduce teacher-forcing as training progresses. Lastly, we choose to expand two candidates per step during training, as it helps reduce cascading errors at test time.

## 3 EXPERIMENTS

**List construction in Lisp**    We focus on nested list construction as a natural starting point towards expressive computation. We test on 100 problems held out from training, and report the percentage of problems solved within 200 steps. The maximum time the RNN-Guided search used was 11 minutes, so we limit the naive (biased interleaving without heuristics) and heuristic searches to 30 minutes.[3] Results are shown in Table 1A.

**Generalization and Comparison**    In a second set of experiments, we use the same model weights as above to demonstrate generalizability. We synthesize three families of programs of varying complexity: `Repeat(N)` which repeats a token $N$ times, `DropLast(N)` which drops the last element in an $N$ element list, and `BringToFront(N)` which brings the last element to the front in an $N$ element list. As a measure of how synthesis difficulty increases with $N$, `Repeat(N)` takes $4 + 3N$ steps, `DropLast(N)` takes $\frac{1}{2}N^2 + \frac{5}{2}N + 1$ steps, and `BringToFront(N)` takes $\frac{1}{2}N^2 + \frac{7}{2}N + 4$ steps. The largest training program takes optimally 22 steps to synthesize. The number of optimal steps in synthesis correlates linearly with program size.

We compare against state-of-the-art systems $\lambda^2$, Escher, and Myth. All three use type information, so we could not compare against them fairly in Table 1A[4]. For `Repeat(N)`, `DropLast(N)` and `BringToFront(N)`, typed systems should have an advantage. Further, $\lambda^2$ assumes advanced language constructs like `fold` that other methods do not, and Escher requires an "oracle" to provide outputs for additional inputs. We limit the number of input/output examples to five, and allow every method up to 30 minutes. Our model is further restricted to 200 steps for consistency with Table 1A[5]. Results are shown in Table 1B. Our method is able to solve problems much larger than those seen in training, and is competitive in its potential to generalize to larger programs.

Table 1: Synthesis Results. A. Test Problems Solved (%); B. Generalization: largest $N$ for which synthesis succeeded, and failure modes (out of **time**, out of **memory**, requires **oracle**, other **error**)

| Method | A. Test | B. Generalization | | |
| --- | --- | --- | --- | --- |
| | % Solved | Repeat(N) | DropLast(N) | BringToFront(N) |
| Naive (Byrd et al., 2012) | 27% | 6 (time) | 2 (time) | - (time) |
| +Heuristics (Byrd et al., 2017) | 82% | 11 (time) | 3 (time) | - (time) |
| RNN-Guided (Ours) | **99%** | **20+** | **6** (time) | **5** (time) |
| $\lambda^2$ (Feser et al., 2015) | | 4 (memory) | 3 (error) | 3 (error) |
| Escher (Albarghouthi et al., 2013) | | 10 (error) | 1 (oracle) | - (oracle) |
| Myth (Osera & Zdancewic, 2015) | | **20+**[6] | - (error) | - (error) |

## 4 CONCLUSION AND FUTURE WORK

We presented a neural guided synthesis model where the neural guide takes as input the internal constraint encoding of the PBE problem used by miniKanren. We show promising results in the model's ability to generalize to larger problems. The ability to encode recursive problems is available in miniKanren, so learning to guide recursive program synthesis is left as future work.

---

[3] All test experiments are run on Intel i7-6700 3.40GHz CPU with 16GB RAM.

[4] Problems in Table 1A are dynamically-typed, improper list construction problems.

[5] If given the full 30 minutes, our model is able to synthesize DropLast(7) and BringToFront(6).

[6] Myth solved Repeat(N) much faster than our model, taking <15ms per problem.

ACKNOWLEDGMENTS

Research reported in this publication was supported in part by the Natural Sciences and Engineering Research Council of Canada, and the National Center For Advancing Translational Sciences of the National Institutes of Health under Award Number OT2TR002517. The content is solely the responsibility of the authors and does not necessarily represent the official views of the funding agencies.

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
