# OpenReview forum: "Leveraging Constraint Logic Programming for Neural Guided Program Synthesis"
_ICLR.cc/2018/Workshop — Accept_

### Official Review · AnonReviewer1 · 2018-03-03
**Difficult for non-experts to follow**

**Rating:** 5
**Confidence:** 1

**Review:**

=== Summary

In this workshop paper, the authors propose to use an LSTM as a heuristic function to guide the search within a constraint logic programming language; the goal of the search is to find (synthesize) a program which reproduces given input-output pairs. Preliminary results suggests that the LSTM-guided search solves larger problems than previous approaches.

=== Comments

As a non-expert in program synthesis, I found this paper extremely difficult to follow. For example, the authors mention expanding a search node, but the text does not describe what “expanding” really means; presumably, it somehow corresponds to adding constraints, but it is not clear.

Further, embeddings found with the LSTM are somehow scored (and then pooled), but the scoring function is not described. Is the scoring function for the embeddings somehow related to the loss on the softmax? or something domain specific?

It is not clear to me why there is a unique optimal candidate for each step in training, or how such a candidate can be found. Is this somehow extracted when generating the programs?

---

As mentioned, I am not an expert in this domain; I am skeptical whether this work would appeal to a broad audience without adding more context. Still, program synthesis-related papers have been presented at previous ICLR conferences, so it may appeal to that sub-community.

---

### Official Review · AnonReviewer3 · 2018-03-09
**Interesting paper, needs more context**

**Rating:** 7
**Confidence:** 4

**Review:**

This paper uses a guided search in a constraint programming language to perform program synthesis. Doing the guided search in a constraint language greatly constrains the search itself, which when combined RNN model to score paths and RL to search yields speedups in generating valid programs.

This is a novel and interesting result which I hope to see extended.

This paper assumes familiarity with minikanren which makes it hard to follow. It would help if the following phrases were better explained, "relational interpreter", "running programs backward", "constraint type". Many of these terms might be known to the logic programming community, but are not known more broadly. It would help to spend a little time on evalo and lookupo just to understand constraint language a bit better.  I'm unsure why minikanren was used vs another logic language. The paper mentions minikanren has the potential to synthesize recursive programs, but none of the experiments suggest that a recursive program is ever generated.

It is a shame there isn't a more direct comparison that can be made between this system and other program synthesis systems. The results are very impressive when compared against systems with type information, and I expect the results could be more dramatic if the comparison were more fair.

Solid paper overall.

---

### Official Review · AnonReviewer2 · 2018-03-10
**RNNs for guiding program search**

**Rating:** 6
**Confidence:** 4

**Review:**

This paper presents a technique for using RNNs to guide program search in a system called miniKaren. The search process involves expanding a series of incomplete partial programs until a complete program is obtained that satisfies the I/O examples. Instead of some of the recent approaches that generate programs directly as output in an end-to-end fashion, the idea here is to guide an already existing search method. The results show significant improvements in search over non-learning based original synthesis solver and also compare favorably to other PL baselines.

Overall, this is an interesting paper and shows the usefulness of combining neural and symbolic search approaches. In terms of modeling, the model to encode and score partial programs is relatively straightforward (LSTMs with pooling). However, the idea of embedding internal solver state (in terms of constraints) is quite interesting.

Although the results show significant improvements, it would be interesting to better understand more qualitatively what kinds of patterns the model is learning. For example, is it the case that the model is learning the manually defined heuristics (likely even more since it performs even better). Some details in the evaluation are also missing. It wasn't clear how long the constraints are that are being embedded and what are the sizes of the test programs in 1A. Are there also some non-synthetic real-world programs for evaluating miniKaren where the improvements can be shown?

---

### Decision · Program_Chairs · 2018-03-20
**ICLR 2018 Workshop Acceptance Decision**

**Decision:**

Accept

**Comment:**

Congratulations, your paper was accepted to the ICLR workshop.